RNN-BiLSTM-CRF based amalgamated deep learning model for electricity theft detection to secure smart grids

Khalid Aqsa 1
Mustafa Ghulam 2 gmustafa@uaar.edu.pk
Rana Muhammad Rizwan Rashid 2
http://orcid.org/0000-0003-0499-7170 Alshahrani Saeed M. 3
http://orcid.org/0009-0008-4830-6590 Alymani Mofadal 4 malymani@su.edu.sa
1 Department of Computer Science, COMSATS University , Islamabad , Pakistan
2 University Institute of Information Technology, PMAS-Arid Agriculture University , Rawalpindi, Punjab , Pakistan
3 Department of Computer Science, College of Computing and Information Technology, Shaqra University , Shaqra , Saudi Arabia
4 Department of Computer Engineering, College of Computing and Information Technology, Shaqra University , Shaqra , Saudi Arabia
Raza Khalid
Electronic publication date: 2024 Feb 26
Publication date: 2024
Volume: 10
Electronic Location ID: e1872
Received 2023 Aug 8; Accepted 2024 Jan 22
Copyright: © 2024 Khalid et al.
Copyright year: 2024
Copyright holder: Khalid et al.
License: This is an open access article distributed under the terms of the Creative Commons Attribution License, which permits unrestricted use, distribution, reproduction and adaptation in any medium and for any purpose provided that it is properly attributed. For attribution, the original author(s), title, publication source (PeerJ Computer Science) and either DOI or URL of the article must be cited.
License URL: https://creativecommons.org/licenses/by/4.0/

Keywords: Smart gird, RNN, BiLSTM, Electricity-theft, CRF

Funding: Deanship of Scientific Research at Shaqra University This work was supported by the Deanship of Scientific Research at Shaqra University. The funders had no role in study design, data collection and analysis, decision to publish, or preparation of the manuscript.

==============================
Electricity theft presents a substantial threat to distributed power networks, leading to non-technical losses (NTLs) that can significantly disrupt grid functionality. As power grids supply centralized electricity to connected consumers, any unauthorized consumption can harm the grids and jeopardize overall power supply quality. Detecting such fraudulent behavior becomes challenging when dealing with extensive data volumes. Smart grids provide a solution by enabling two-way electricity flow, thereby facilitating the detection, analysis, and implementation of new measures to address data flow issues. The key objective is to provide a deep learning-based amalgamated model to detect electricity theft and secure the smart grid. This research introduces an innovative approach to overcome the limitations of current electricity theft detection systems, which predominantly rely on analyzing one-dimensional (1-D) electric data. These approaches often exhibit insufficient accuracy when identifying instances of theft. To address this challenge, the article proposes an ensemble model known as the RNN-BiLSTM-CRF model. This model amalgamates the strengths of recurrent neural network (RNN) and bidirectional long short-term memory (BiLSTM) architectures. Notably, the proposed model harnesses both one-dimensional (1-D) and two-dimensional (2-D) electricity consumption data, thereby enhancing the effectiveness of the theft detection process. The experimental results showcase an impressive accuracy rate of 93.05% in detecting electricity theft, surpassing the performance of existing models in this domain.

Introduction

Electricity constitutes an indispensable component of our daily existence, as it is harnessed, transmitted, and distributed across expansive power grids (Butt, Zulqarnain & Butt, 2021). These grids, strategically positioned near energy sources, undertake the conversion of various forms of energy into electricity, subsequently disseminating it through an intricate network. Nevertheless, power grids are susceptible to illicit attacks, particularly in the form of electricity theft, which detrimentally affects the quality of the electrical supply, leads to power dissipation, and engenders voltage irregularities. Electricity losses can manifest in two primary manners: technical losses (TLs) and non-technical losses (NTLs). Technical losses generally arise from hardware complications within the electrical system, while non-technical losses stem from abnormal data flow within the system. Aberrant electrical flow transpires when electric meters are surreptitiously bypassed, tampered with, or manipulated to yield falsified readings (Saeed et al., 2022). These fraudulent readings result in unrecorded revenues, thereby exerting an adverse impact on the economy. For instance, reports indicate that electricity theft in Canada has incurred losses amounting to approximately $100 million.

The concept of the smart grid (SG) focuses on the continual expansion of renewable and distributed power sources, aiming to achieve flexibility, self-healing capabilities, efficiency, and sustainability (Ullah et al., 2022). This innovative idea is gaining increasing recognition as it combines advanced infrastructure with existing power grids (Guerrero et al., 2014). The employed cyber infrastructure facilitates the collection and analysis of data from various distributed endpoints, such as phase or measurement units, smart meters, and circuit breakers. The architecture of the smart grid is illustrated in Fig. 1. The introduction of advanced meter infrastructure (AMI) has introduced a new dimension to conventional power grids (Hasan et al., 2019). AMI incorporates smart meters, sensing devices, computational equipment, and modern communication technologies, enabling seamless bidirectional communication between consumers and utilities. Its primary objective is to gather crucial data related to energy consumption (EC), prevailing prices, and power grid status. However, the integration of the Internet into AMI also poses potential security vulnerabilities, as it creates opportunities for malicious actors to remotely exploit the intelligent meter infrastructure and engage in unauthorized electricity theft.

Figure 1 A typical architecture of smart grid.

Various conventional methods are available for detecting power theft, such as comparing abnormal meter readings with normal ones, meter re-installation, and looking for meter misconfigurations. However, applying these methods to large volumes of data is challenging, and they often produce inaccurate results. Smart grids, which are advanced electric networks, present significant opportunities for detecting electricity theft. Smart grids operate on the principle of edge computing. They possess self-healing capabilities and enable the identification of malicious data flow through detection-reaction mechanisms. Previous studies (Hussain et al., 2021; Wang et al., 2017) have demonstrated the effectiveness of smart grids in detecting electricity theft.

Therefore, this research article aims to overcome these limitations by proposing an ensemble method for electricity theft detection based on a combination of recurrent neural network (RNN) and bidirectional long short-term memory (BiLSTM). RNNs are deep neural networks commonly utilized for image classification tasks, offering the advantage of automatically extracting crucial features from data without human intervention. The proposed RNN-BiLSTM-CRF model comprises input layers, hidden layers, and an output layer. To enhance accuracy, the BiLSTM algorithm, which is known for its speed, ease of use, and efficiency, is integrated with the output layer of the RNN model.

The contributions of the proposed model can be succinctly summarized as follows: Introduction of an ensemble model, RNN-BLSTM-CRF, for electricity theft detection, combining a Recurrent neural network and a Bidirectional Long Short-Term Memory.

Utilization of both daily and weekly electricity consumption data, with the wide component incorporating daily data and the deep component incorporating weekly data.

Comparison with other classification models, demonstrating the superior accuracy of the proposed model.

The remainder of the article is structured in the following manner: The study is broken up into five sections: “Literature Review”, which gives a review of relevant literature; “Materials and Methods”, which demonstrates the framework of the proposed model; “Results”, which includes experimental data; and “Conclusion”, which concludes the research.

Literature review

In this section, a brief survey on electricity-theft detection is presented. Power misappropriation stands as a key non-technical liability within decentralized networks, inflicting significant damage upon power grids (Khan et al., 2020). As these grids furnish centralized power to connected consumers, any deceitful usage can undermine the grids, disrupting the entire power supply and compromising its quality. Detecting such fraudulent consumers becomes arduous in the face of extensive data. Smart grids offer a viable solution by facilitating bidirectional electricity flow, enabling the identification, replication, and implementation of new modifications to the electrical data stream. Traditional systems for detecting power theft rely on one-dimensional (1-D) electrical data, yielding subpar accuracy in theft identification. To address this issue, this article introduces an ensemble model “CNN-XGB”. The proposed model attains a superior 92% accuracy in detecting electricity theft, surpassing existing models.

Electricity deception, a significant portion of non-technical losses, has remained a pressing global issue. However, upon detecting fraud, a costly and labor-intensive on-site inspection becomes essential for final validation (Emadaleslami, Haghifam & Zangiabadi, 2023). Given the challenges of inspecting all metering data, utilities continuously strive to narrow down inspections to instances with higher fraud probabilities. Smart meters have ushered in AI-driven approaches to minimize inspection scope by analyzing consumption patterns. Nevertheless, their effectiveness is constrained by data imbalance, resulting in insufficient malicious samples. Using metering data, a two-stage deep-learning algorithm detects energy theft. First, a clever architecture detects fraudulent customers’ suspicious behavior, which helps manage data imbalance by forecasting regular consumers’ theft situations. Next, a deep neural network (DNN) network distinguishes normal from suspicious customers. Clustering, deep learning, and consumption and lifestyle data allow the system to understand client behavior without being mislead by non-malicious consumption patterns. It performed well on 6,000 real-world metering data.

Power distribution firms face income loss from non-technical energy theft losses. Smart grids provide bidirectional energy management, invoicing, and load monitoring in real time. Data-driven solutions allow power distribution companies to automate energy theft detection (ETD) using this infrastructure (Xia & Wang, 2022). Due to high-dimensional unbalanced data, feature extraction loss, and expert participation, standard ETD algorithms fail to identify. Thus, this article proposes an RDAE-AG-TripleGAN semi-supervised solution for ETD. The RDAE retrieves features and associations, while the AG component weights features and dynamically monitors the AG-TripleGAN. Thus, this combined method improves ETD. Real power usage data from smart meters was used to simulate the suggested method’s advantages over traditional techniques. Simulations show that the suggested technology detects electricity theft better than existing research. The suggested solution gives electric utilities a more reliable and acceptable technology with a detection rate of 0.956.

Electricity theft is a prevalent issue across nations, causing detrimental effects on economies. Detecting theft within the power sector poses a considerable challenge for power distribution entities, as it leads to financial losses and energy wastage (Nawaz et al., 2023). Due to the large scale of these incidents, manual examination of each case is labor-intensive. Therefore, automated detection of electricity theft has become an urgent necessity. This study introduces a model for electricity theft detection that utilizes data from smart grid meters, leveraging the power of eXtreme Gradient Boosting (XGBoost) and optical character recognition (OCR). The model incorporates feature selection techniques to identify the most relevant attributes from the dataset for our electricity theft detection model. XGBoost is particularly effective due to its regularized model formulation, which mitigates overfitting and improves overall performance at a faster rate. Additionally, OCR is employed to determine the specific objects associated with instances of electricity theft. By employing XGBoost and OCR, this approach presents a highly efficient method for identifying cases of electricity fraud, enabling power distribution organizations to curb theft effectively and minimize financial losses.

The emergence of Industry 4.0 marks a new era of industrial revolution, characterized by the interconnectedness of machines and activities through network sensors, generating vast amounts of data (Karimipour et al., 2019). Machine learning (ML) and deep learning (DL) plays pivotal role in analyzing this data to extract valuable insights for manufacturing enterprises, thereby paving the way for the development of Industrial AI (IAI). This article sheds light on the fundamental principles of Industry 4.0, highlighting its distinctive features, requirements, and challenges. Furthermore, a novel architecture for IAI is proposed, considering the unique demands of this paradigm. Moreover, this study delves into the essential ML and DL algorithms that find applications in Industry 4.0. Each algorithm is meticulously examined, considering its specific characteristics, diverse applications, and overall efficiency. Subsequently, the focus shifts to the smart grid domain, one of the most critical sectors within Industry 4.0. ML and DL models designed for smart grid applications are presented and thoroughly analyzed in terms of their effectiveness and efficiency. Lastly, this article addresses the emerging trends and challenges in data analysis within the context of the new industrial era. Key aspects such as scalability and cyber security are discussed, underscoring the significance of robust solutions to overcome these obstacles and ensure the smooth operation of advanced manufacturing systems.

Electricity theft stands as a significant contributor to power loss, impacting both the quantity and quality of electrical supply. However, the existing approaches for detecting such illicit activities present a range of complexities, primarily due to the inherent irregularities within the time-series data (Hussain et al., 2021). In light of the increasing importance of efficient energy utilization, particularly in the context of electricity, power theft has emerged as a grave concern. The power system comprises three components: generation, transmission, and distribution. Power theft primarily occurs at the transmission and distribution stages. Numerous strategies have been proposed to combat electricity theft, yet investigating direct connections to the distribution lines poses significant challenges. To address this issue, we propose leveraging various machine learning techniques to effectively detect instances of electricity theft.

Materials and Methods

Our methodology consists of the following three stages: (1) data pre-processing. (2) Data balancing, (3) extraction of features with the RNN model, and (4) classification utilizing the BiLSTM-CRF model. The proposed model utilizes both one-dimensional (1-D) and two-dimensional (2-D) electricity consumption data. The proposed model is shown in Fig. 2.

Figure 2 Proposed model.

Data collection

This research study utilized three distinct datasets. The first dataset was obtained from the State Grid Corporation of China (SGCC) and consisted of real power usage data from customers (Yi-Chong, 2018). The dataset was presented in a numerical format and included the customer ID, the days of the week, Land Classification and the quantities of energy consumed by each customer over a 2-year period. It also included labeled classes indicating whether the users were normal users (8,562) or theft users (1,394). The dataset also provided numerical information for any missing or faulty data. The second dataset was obtained from Haq et al. (2023) through a public request. It contained records of energy usage from 560,640 industrial and residential customers. The dataset included one normal class and six different theft classes, with varying numbers of instances in each class. The data was collected over a 10-min interval using smart meters from May 2019 to April 2020.The third dataset was collected by Badr et al. (2023) from the Ausgrid dataset and the SOLCAST website. The Ausgrid dataset, provided by Ausgrid, the largest electricity distributor along the eastern coast of Australia, consisted of real measurements of power usage and generation. The data was recorded every 30 min by a group of consumers located in Sydney and New South Wales, Australia. These consumers had solar panels installed on their rooftops. The readings covered the time period from 1 July 2021 to 30 June 2022. These datasets includes both one-dimensional and two-dimensional electricity consumption data. Table 1 provides statistical information for each of the three datasets mentioned above.

Table 1 Dataset description.

S. No.	Data generator	Nature of dataset	Web link	
SMD-I	State grid corporation of China (SGCC)	Smart meter reading	http://www.sx.sgcc.com.cn/sgcc/index.html	
SMD-II	Local dataset	Customer electricity consumption	https://data.mendeley.com/datasets/c3c7329tjj/1	
SMD-III	Electricity on Australia’s East Coast (AUSGRID)	Customer electricity consumption	https://www.ausgrid.com.au	

Data preprocessing

Electricity consumption data is typically extensive, but it often contains noise and missing values due to the susceptibility of smart meters (Liu et al., 2021). Consequently, it is necessary to clean the data before conducting analysis. To address the issue of missing values, an interpolation method is employed. This method operates on the principle of arithmetic mean, whereby the missing value is replaced with the average of its preceding and succeeding values. The following is the equation that should be used for the interpolation technique:

(1) I(yj)=∑▒〖((yj−1+yj+1)/2)〗

where yj represents the value that is absent, yj−1 represents the value that came before it, and yj+1 represents the value that will come after it. In addition, we used an empirical rule to recover the erroneous data, which is known as the three-sigma rule. This rule makes use of three different standard deviation methodologies to recover the erroneous data.

(2) S(yj)=avg(y)+2⋅std(y);

∀yj>avg(y)+2⋅std(y)

where yj stands for the incorrect value, std(y) represents the standard deviation of the data, and avg(y) represents the average of the information that was used.

As a result, when we had finished removing any outliers and cleaning up the noisy data, we continued to normalize the data that was left by using the MAX-MIN scaling approach. This method makes use of a linear transformation methodology, in which the data are converted based on the extraction of the highest and lowest values from the transformed dataset. This technique is called “linear transformation.” After that, the predetermined MAX-MIN mathematical equation is applied to each and every one of the dataset’s values in order to make a substitution.

(3) M(yj)=(yj−min(y))/(max(y)−min(y))

where min(y) indicates the absolute least value and max(y) indicates the absolute greatest value.

Data balancing

In real-world scenarios, it is commonly observed that the number of regular electricity customers is higher compared to those with unusual consumption patterns. This leads to imbalanced data distribution, which poses challenges for deep learning models in terms of efficiency. The problem becomes more pronounced as it introduces bias towards the larger class in classification models. To address this issue, a technique called SMOTE (Synthetic Minority Oversampling Technique) can be employed. SMOTE generates new instances by modifying existing data points using a synthetic oversampling approach for the minority class (Chawla et al., 2002). The method identifies neighboring data points in the up-sampled data and calculates their distance from each other. By adding a random number, denoted as k, to this distance, the sample is expanded to create synthetic examples. This results in a more balanced distribution of generated features, which can be effectively utilized in the system layer. The algorithm for data balancing using SMOTE is shown in Algorithm 1.

Algorithm 1 Data balancing using SMOTE.

Variables   Majority Factors F+, Minority Factor  F−, threshold h^, ratio R^, Euclidean distance  ∂,	
     Generated Samples S	
Input:  Total number of majority factors F+ and minority Factors F+	
1.    Set threshold 〖h^〗th -> max(degrees(classimbalance))	
2.    ȒFor every minority factor f, calculate Euclidean distance ∂	
3.     Ȓi=Δi/k,k=10	
4:     R^f<R^i/∑▒R^i	
5:     β=F+/F−	
6:     S=(F+−F−)∗β	
Output:  No of S	

Feature extraction

The feature extraction process is crucial to the accurate classification. In this work the services of well know algorithm recurrent neural network (RNNA) recurrent neural network (RNN) is a sort of artificial neural network that enables the output of particular nodes to impact future input to those same nodes by virtue of the fact that the connections between the nodes create a cycle (Zhu et al., 2022). This characteristic allows RNNs to display temporal dynamics, meaning they can effectively handle sequences of inputs with varying lengths by utilizing their internal state, or memory. As a result, RNNs find utility in tasks such as extracting features.

RNNs, in contrast to deep neural networks, incorporate hidden states as learning parameters (Sridhar et al., 2022). These hidden states play a crucial role in processing sequential data. During training, the hidden states are kept up to date by taking into account the information that came before as well as the information that is now available from the sequential data.

This phase involves extracting features from both one-dimensional (1-D) time-series data and two-dimensional (2-D) spatial data. For the 1-D aspect, we focus on extracting time series features. In the case of 2-D data, we concentrate on attributes such as Population Density, Land Classification (Urban/Rural), and Environmental Conditions. Additionally, we incorporate combined features, specifically Energy Efficiency Ratings and Peak Demand Times, to enhance the feature extraction process. These meticulously extracted features are pivotal in enabling our model to proficiently handle and analyze both one-dimensional (1-D) and two-dimensional (2-D) aspects of electricity consumption data.

The hidden state and the output yt of the recurrent layer, where t∈{1,…,T} denotes the index of the frame, are calculated as follows:

The hidden state, denoted by ht, and the output of the recurrent layer, denoted by yt, are computed in the following manner, where t represents the index of the frame and t∈{1,…,T}.

The hidden state, denoted by ht, and the output of the recurrent layer, denoted by yt, are computed in the following manner, where t∈{1,…,T} respectively:

(4) ht=tanh⁡(Whhht−1+Wxhxt+bh)

(5) yt=Whyht+by

where W and b, respectively, describe the weights and biases that are applied between the items.

The current input xt, as well as the prior hidden state, ht−1, are both used in the process of updating the hidden state ht. Therefore, given sequential data, the information that came before it has an effect on the neural network’s computation of the information that comes after it. Through the use of Eq. (2), the output values of the recurrent layer may be determined.

BiLSTM-CRF framework

Long short-term memory is an example of recurrent neural network (RNN) architecture, which was developed in order to solve the issue of disappearing gradients (Zha et al., 2022). This issue was the motivation for its creation. Because LSTMs are able to learn and remember information across extended sequences, they are well suited for tasks that include sequential or time-series data. Some examples of such tasks include voice recognition, natural language processing, machine translation, and sentiment analysis. Memory cells are an essential component of LSTM networks. These cells enable the network to retain information and retrieve it even after it has been dormant for a significant amount of time. These memory cells are made up of a cell state and three different kinds of gates, which are the input gate, the forget gate, and the output gate respectively. The entry and exit of information into and from the memory cells is controlled by these gates. LSTM has the benefit of selectively reading and writing information, which helps it considerably compensate for the shortcomings of gradient explosion and gradient disappearance. Although LSTM is effective at solving the issue of long-term dependencies, it is difficult to make use of the contextual information provided by the text. The fundamental idea behind the construction of the BiLSTM model is that the feature data collected at time t should simultaneously include information about both the past and the future (Park et al., 2023). Both the forward LSTM and the reverse LSTM have the capability of acquiring the aforementioned information on the input sequence. In order to retrieve the complete representation of the hidden layer, the context information of the input sequence is first computed, and then vector splicing is used. It is important to point out that the parameters of the LSTM neural network used in BiLSTM are not reliant on one another in any way; the only thing they have in common is the word vector list used for word embedding (Wu et al., 2023).

By processing the input sequence in both the forward and the backward direction, BiLSTMs are able to accomplish their purpose of capturing contextual information. They are able to describe relationships between neighboring tokens in a sequence as a result of this. They do not, however, clearly describe global dependencies for the whole of the sequence. This restriction is addressed by the CRF layer, which takes into account the joint probability of all potential label sequences (Meng et al., 2022). Additionally, it takes into consideration the interactions that occur between labels that are located in various places. Because of this, the model is able to account for long-range relationships and provide forecasts that are more grounded in reality.

Results

An extensive analysis of the findings is provided in this section. A thorough series of experiments was run in order to gauge the proposed system’s accuracy and utility. With the help of three distinct benchmark datasets, all of which were drawn from previously published research, the functionality of the proposed system was assessed. The experimental analysis’s findings demonstrated that the performance of the suggested strategy outperformed other cutting-edge methods currently in use.

Baseline method

Using the data sets displayed in Table 2, we evaluate the performance of the proposed model by comparing it to the baseline models provided below. Baseline 1: Ullah et al. (2022) proposed a technique that were based on AdaBoost, AlexNet and Artificial Bee models.

Baseline 2: Ahir & Chakraborty (2022) proposed a technique for detection of electricity theft using context-aware approach and pattern based approach.

Baseline 3: Hasan et al. (2019) presented a method based on deep CNN and LSTM computations for the detection of power theft in smart grids.

Table 2 Performance imbalanced data and balanced data.

Parameters	Applying on imbalance data	Applying on balanced data	
DS1	DS2	DS3	DS1	DS2	DS3	
Precision	Non-malicious user	0.92	0.91	0.90	0.90	0.91	0.89	
Malicious user	0.62	0.64	0.63	0.87	0.89	0.69	
Recall	Non-malicious user	0.96	0.98	0.97	0.87	0.89	0.93	
Malicious user	0.45	0.44	0.45	0.91	0.92	0.48	
F1-score	Non-malicious user	0.94	0.96	0.98	0.89	0.91	0.93	
Malicious user	0.52	0.58	0.54	0.89	0.94	0.51	
Overall accuracy	0.89	0.89	0.88	0.89	0.88	0.92	

Performance matrices

The effectiveness of the suggested strategy is assessed using three different types of performance criteria.

F-measure

The suggested model was evaluated using classification measures such as accuracy, F1 score, precision, and recall. The formulations for each of these measurements are given in Eqs. (6)–(9).

(6) Accuracy=sum(TruePositives,TrueNegatives)sum(TrueNegatives,FalsePositives,TruePositives,FalseNegatives)

(7) F1=2∗Precision∗RecallPrecision+Recall

(8) Precision=count(TruePositives)sum(TruePositives,FalsePositives)

(9) Recall=count(TruePositives)sum(TruePositives,FalseNegatives)

Matthews correlation coefficient (MCC)

MCC is mostly used to check performance of a problem with classification of binary nature. The MCC, which is a single integer, can be extracted from the parameters of a confusion matrix. Calculating MCC is shown in Eq. (10).

(10) MCC=(TP∗TN−FP∗FN)(TP+FP)(TP+FN)(TN+FP)(TN+FN)

The MCC range, which spans from −1 to +1, represents the accuracy of the binary classification model in correctly predicting positive and negative cases while accounting for the imbalance between the two classes. A flawless prediction is indicated by an MCC score of 1, or −1. The MCC cannot correctly discriminate between positive and negative examples if it receives a score of 0. A negative MCC score of −1, however, indicates that the model is behaving very ineffectively.

Binary cross-entropy

The loss function known as Binary Cross Entropy (BCE) is often used in binary classification problems. It determines the discrepancy between the expected probability distribution by Qi and the true binary labels represented by Pi. Equations (11) and (12) provide the formula for computing the cross entropy.

(11) H(Pi,Qi)=∑iPilogQi

The cross-entropy decreases and approaches zero as the forecast gets closer to being accurate. Only two probabilities are utilized in the loss computation when the classification model is only used to classify two classes. It has the following mathematical definition:

(12) BCE=PilogQi−(1−Pi)log(1−Qi).

BCE compares the model’s predicted probabilities with the actual binary labels to calculate the average dissimilarity or correctness.

Experimentation results

By evaluating the precision, accuracy, and recall of the suggested method for electricity theft detection, the first experiment demonstrates its effectiveness. It can be witnessed that proposed technique scored highly across all datasets, as shown in Fig. 3, indicating outstanding results in terms of recall, precision, and accuracy for each dataset.

Figure 3 Experimental results on DS-1, DS-2 and DS-3.

In comparison to testing the algorithm on synthetic data, the model was unable to effectively classify the fraud users when unbalanced data was supplied to it. Table 2 provides information on performance metrics for both imbalanced and synthetic data. The suggested technique scored highly across all datasets, as shown in Fig. 3, indicating outstanding results in terms of recall, precision, and accuracy for each dataset.

Another widely used statistic for binary classification is the MCC, which we utilize in this experiment. Figure 4 shows three more performance measures in relation to the number of epochs. While the validation loss grew and accuracy showed a declining trend, the train loss and accuracy increased as the number of epochs increased. The model overfits in this situation because it was not generalized for the unknowable input.

Figure 4 ROC, confusion matrix of malicious and non-malicious user on balanced dataset.

When the datasets in question are unbalanced, the MCC is frequently used. The accuracy figures for regular users and theft users who do not employ SMOTE are incongruent with one another, as seen in Table 2. Despite the fact that overall accuracy was essentially the same before and after SMOTE was adopted, this is the case. This is due to the fact that, despite the model’s high degree of accuracy, the test dataset’s uneven distribution hinders the model from correctly identifying the data. The model is then trained using the newly created dataset after the SMOTE has been set up to produce synthetic data. Figure 4 illustrates the training results using the balanced dataset. The train set and the test set do not contain any diverging patterns for patterns that are identical to one another. After the first 500 epochs, the test’s accuracy pattern remained consistent, and after the first 600 epochs, it had an accuracy rate of 89%. The test loss also appeared to be unchanged after the same number of epochs. The test set’s MCC also yielded a value of 0.82, indicating that the model is capable of making precise predictions.

Comparison to existing models

According to what was previously stated in the baseline techniques sections, the performance of the proposed model was compared to baseline models.

Table 3, gives insight of the performance comparison of a proposed technique with a baseline technique (Ullah et al., 2022; Ahir & Chakraborty, 2022; Hasan et al., 2019). The proposed technique attained a precision of 90.02%, recall of 84.18%. The F-score was calculated to be 86.62%. The accuracy of our proposed model was 93.05%. The Baseline one model, on the other hand gained an accuracy of 87.94%, a precision of 86.05%, a recall of 84.21%, and an F-score of 83.20%. The improved results show the effectiveness of proposed model in terms of used measures.

Table 3 Comparison between proposed model and baselines.

Technique	Precision	Recall	F-score	Accuracy	
Proposed model	90.02	84.18	86.62	93.05	
Baseline 1	86.05	84.21	83.20	87.94	
Baseline 2	89.24	83.23	83.90	90.13	
Baseline 3	80.82	76.23	77.15	79.49	

The proposed module utilizes an amalgamated model based on deep learning, which is achieved by combining the architectures of bidirectional long short-term memory (BiLSTM) and recurrent neural networks (RNN). The astute amalgamation of RNN and BiLSTM components in this model improves its capacity to comprehend contextual details and sequential dependencies present in the data. Recurrent neural networks (RNNs) are highly suitable for processing sequential data, whereas BiLSTM networks, by virtue of their bidirectional architecture, demonstrate exceptional proficiency in capturing information from both past and future contexts. By amalgamating these two formidable components, our suggested module capitalizes on the respective merits of each architecture, thereby producing an all-encompassing and efficient framework for the designated application. The performance of the module is enhanced by this novel combination, which distinguishes it from traditional models that depend on a solitary architecture.

As demonstrated in Fig. 5, the proposed model performs better than the baseline models (Ullah et al., 2022; Ahir & Chakraborty, 2022; Hasan et al., 2019) in terms of precision, recall, F-score, and accuracy. These findings suggest that the suggested approach has promise and might work for the current project. However, more research and testing are required to confirm its advantages and establish its practicality for practical applications.

Figure 5 Comparsion with baseline approaches.

Conclusions

A prevalent non-technical issue that harms both power infrastructure and electricity users is electricity theft. It has an impact on consumers’ high energy costs, restricts utility companies’ capacity to grow economically, and raises electrical safety issues. The study addresses the significant issue of electricity theft and proposes a novel ensemble model, RNN-BiLSTM-CRF, leveraging recurrent neural network (RNN) and bidirectional long short-term memory (BiLSTM) architectures. Traditional methods face challenges, but the proposed model achieves a notable 93.05% accuracy, outperforming existing approaches. This innovation, applied to smart grid data, demonstrates the potential for effective machine learning solutions in mitigating electricity theft’s economic and safety impacts. The current model’s training is based on a restricted set of datasets and does not take into account additional non-sequential parameters. This limitation hampers its ability to accurately identify instances of theft. Furthermore, the inclusion of inadequately sampled data affects the performance of the proposed model when it comes to obtaining more detailed insights into energy theft. To ensure a dependable detection of energy thieves in the future, we will therefore prioritize the utilization of high-sampling data and explore the incorporation of non-sequential data.

Supplemental Information

Supplemental Information 1 Model Code.

This model is developed in Python.

Additional Information and Declarations

Competing Interests

Author Contributions

Data Availability

The authors declare that they have no competing interests.

Aqsa Khalid conceived and designed the experiments, prepared figures and/or tables, and approved the final draft.

Ghulam Mustafa conceived and designed the experiments, performed the experiments, prepared figures and/or tables, authored or reviewed drafts of the article, and approved the final draft.

Muhammad Rizwan Rashid Rana performed the experiments, performed the computation work, authored or reviewed drafts of the article, and approved the final draft.

Saeed M. Alshahrani analyzed the data, prepared figures and/or tables, and approved the final draft.

Mofadal Alymani analyzed the data, authored or reviewed drafts of the article, and approved the final draft.

The following information was supplied regarding data availability:

The proposed model code is available in the Supplemental File.

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
