# Peer review of "RNN-BiLSTM-CRF based amalgamated deep learning model for electricity theft detection to secure smart grids"

_PeerJ Computer Science, doi:10.7717/peerj-cs.1872_

## Round 0.1 · original submission · Major Revisions

Reviewers found merit in the paper, however, they recommended significant revision. Authors are required to revise the manuscript and address all the reviewers' comments.

**Language Note:** The review process has identified that the English language must be improved. PeerJ can provide language editing services - please contact us at copyediting@peerj.com for pricing (be sure to provide your manuscript number and title). Alternatively, you should make your own arrangements to improve the language quality and provide details in your response letter. – PeerJ Staff

·

Basic reporting

No Comment.

Experimental design

No comment.

Validity of the findings

No comment.

Reviewer 2 ·

Basic reporting

The paper titled "RNN-BiLSTM-CRF based amalgamated deep learning model for electricity theft detection to secure smart grid", is a novel writing. The proposed model used both one-dimensional (1-D) and two-dimensional (2-D) electricity consumption data, to enhance the effectiveness of the theft detection process is very impressive. The paper is clearly written in a good style and includes figures and tables wherever necessary.

Experimental design

The objective and motivation for the research has been very well stated in the introduction part. But needs clarification on the following:
1. These objective need more clarification
2.The authors describes SMOTE algorithms for data balancing, but regarding the chosen datasets which parameter are chosen and what was the results not mentioned properly. So needs proper clarifications.
3. Use of evaluation measures Matthews Correlation Coefficient (MCC) and Binary Cross-Entropy to prove the effectiveness of the proposed method is quite satisfactory. The authors have clearly acknowledged and identified the contributions of their research against previous researchers' work.

Validity of the findings

The authors adequately evaluated their work, and all claims are clearly articulated and supported by empirical experiments.

Additional comments

The manuscripts has some typo and grammatical errors like :"RNN-BiLSTM-CRF based amalgamated deep learning model for electricity theft detection to secure smart gird" title has written gird instead of grid.
However, addressing the above comments would improve the quality of the paper. The overall work is good, novel and timely.

·

Basic reporting

I found the subject of study quite fascinating. It is quite striking. Congratulations to the researchers. However, the study needs to be improved in terms of the following issues.
1- There is unnecessary information in the abstract, there is nothing necessary. We expect a scientific manuscript presenting the results of the study, not a report summarizing a study. I think we should see the following information in a sentence in the abstract.
The title spelled wrong-Smart Grid. Authors have written smart gird.
1.1. What was done in this study?
1.2. Why was this study done?
1.3. What results were obtained in this study?
1.4. To what extent is the obtained result successful compared to the equivalent studies?
The abstract section should answer the questions above. Especially the success performance in the last item should be given.

Experimental design

2-In the conclusion section, the summary section should not be repeated, but the results of this study and the success achieved should be summarized in just a few sentences.
3-The success achieved in the study and the reasons why this success was achieved should be explained in detail in a paragraph. Its superiority over similar studies must be demonstrated.
4-The organization and structure of the article should be revised to make the subject more understandable.

Validity of the findings

5-I think that Result, Discussion and Conclusion sections should exist separately for such an article.
6-Performance criteria that show the success of the work should be emphasized more clearly.
6-Authors are advised to add latest citations.

Additional comments

7-Current references on the subject should be added. The number of references is not small. However, in this type of study, I think there should be much more up-to-date references.
8-Overall, there are still some minor parts that the authors did not explain clearly. Some additional evaluations are expected to be in the manuscript as well. As a result, I am going to suggest Minor revision of the paper in its present form.

·

Basic reporting

1. The title is RNN-BiLSTM-CRF but in the discussion, it is mentioned as CF ??

2. The article is written according to professional standards.

3. Figure 1 and Figure 2 don't give a clear understanding.

Experimental design

1. The article is in the scope of the journal

2. the proposed model leverages both one-dimensional (1-D) and two-dimensional (2-D) electricity
consumption data, But it has not been discussed in the methodology.

3. Recurrent Neural Network (RNN) and Bidirectional Long Short-Term Memory (BiLSTM) architectures combination is not quite novel. What moderations are introduced to enhance the proposed model performance?

Validity of the findings

1. The baseline methods used the same 3 datasets that the author has used? If not then the authors should compare existing methods evaluated on the same dataset.

2. More detailed experimental analyses are required

Additional comments

1. The manuscript is well structured

2. List the major contributions of your work

3. Extensive comparative analysis is required

---

## Round 0.2 · Minor Revisions

The quality of the manuscript has been significantly improved. However, one of the reviewers is still concerned about the description of the datasets. Still, the one-dimensional (1-D) and two-dimensional (2-D) electricity consumption data has not been discussed in the methodology. Kindly address this issue and resubmit.

Reviewer 2 ·

Basic reporting

The proposed model used both one-dimensional (1-D) and two-dimensional (2-D) electricity consumption data, to enhance the effectiveness of the theft detection process is very impressive. The paper is clearly written in a good style and includes figures and tables wherever necessary.

Experimental design

The objective and motivation for the research has been very well stated and all the suggestions are incorporated in required sections.
Use of evaluation measures Matthews Correlation Coefficient (MCC) and Binary Cross-Entropy to prove the effectiveness of the proposed method is quite satisfactory. The authors have clearly acknowledged and identified the contributions of their research against previous researchers' work.

Validity of the findings

The authors adequately evaluated their work, and all claims are clearly articulated and supported by empirical experiments.

Additional comments

The overall work is good, novel and timely.

·

Basic reporting

The article is written according to professional standards.

Experimental design

Still, the one-dimensional (1-D) and two-dimensional (2-D) electricity consumption data has not been discussed in the methodology.

Validity of the findings

comparison to existing methods is ok....

Additional comments

The manuscript is well-structured

still Extensive comparative analysis is required..

---

## Round 0.3 · accepted · Accept

The manuscript has been revised as per the reviewers' and editor's suggestions. It may be accepted for publication in the current form.